# Improved Twin Delayed Deep Deterministic Policy Gradient Algorithm Based Real-Time Trajectory Planning for Parafoil under Complicated Constraints

**Jiaming Yu [1]** , **Hao Sun [2],\*** and **Junqing Sun [1]**

[1] School of Computer Science and Engineering, Tianjin University of Technology, Tianjin 300384, China
[2] College of Artificial Intelligence, Nankai University, Tianjin 300350, China
* Correspondence: sunh@nankai.edu.cn; Tel.: +86-18502256348

**Abstract:** A parafoil delivery system has usually been used in the fields of military and civilian airdrop supply and aircraft recovery in recent years. However, since the altitude of the unpowered parafoil is monotonically decreasing, it is limited by the initial flight altitude. Thus, combining the multiple constraints, such as the ground obstacle avoidance and flight time, it puts forward a more stringent standard for the real-time performance of trajectory planning of the parafoil delivery system. Thus, to enhance the real-time performance, we propose a new parafoil trajectory planning method based on an improved twin delayed deep deterministic policy gradient. In this method, by pre-evaluating the value of the action, a scale of noise will be dynamically selected for improving the globality and randomness, especially for the actions with a low value. Furthermore, not like the traditional numerical computation algorithm, by building the planning model in advance, the deep reinforcement learning method does not recalculate the optimal flight trajectory of the system when the parafoil delivery system is launched at different initial positions. In this condition, the trajectory planning method of deep reinforcement learning has greatly improved in real-time performance. Finally, several groups of simulation data show that the trajectory planning theory in this paper is feasible and correct. Compared with the traditional twin delayed deep deterministic policy gradient and deep deterministic policy gradient, the landing accuracy and success rate of the proposed method are improved greatly.

**Keywords:** parafoil delivery system; trajectory planning; homing control; twin delayed deep deterministic policy gradient

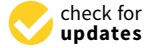



## 1. Introduction

A parafoil delivery system is a special precision air conveying system [1–4]. By controlling the shape of the parafoil canopy, this system can change the flight direction and achieve precise landing, which is hard to realize by the traditional parachute system. Based on this advantage, parafoil has broad prospects in the fields of military and civilian airdrop supply and aircraft recovery. For example, NASA applied a 689 m$^2$ parafoil to recovery X-38 aircraft [5,6]. The German Aerospace Center (DLR) has also developed a small and widely instrumented aircraft Alex (a parafoil delivery system), and proposed the application of the GNC algorithm in autonomous landing [7,8]. Other research, such as [9,10], explored the modeling approach for ram-air parachutes. Ref. [11] proposed pods that provide distributed sensors in the whole parachute canopy and a fusion algorithm to merge the pod data into useful canopy state estimation, which improves the accuracy of canopy state estimation. Ref. [12] presented a feasibility study on the engine and engine frame recovery system of an existing expendable heavy launch vehicle.

By analyzing the dynamic features of parafoil, it can be observed that its vertical velocity is nearly uncontrollable for the unpowered parafoil. Due to this feature, the flight time of parafoil trajectory planning depends mainly on the initial launching altitude. The

initial altitude of the parafoil determines that the flight time and the adjustment time for trajectory change in the flight process is very limited. Parafoil has two research directions of trajectory planning: the optimal trajectory planning and the multiple trajectory planning. Based on the above two research directions, the trajectory planning of the parafoil delivery system is studied by many scholars. The reference flight trajectory is usually composed of standard straight lines and arcs. In [7], a GNC algorithm based on T-Approach is applied to multi-trajectory planning. Meanwhile, the trajectory of optimal trajectory planning is an irregular curve. In the early research on the multiple planning method, Ref. [13] proposed a simplified 6-DOF model predictive control strategy for autonomous control of parafoil. The trajectory of this method is composed by many short straight lines. Then, Refs. [14–16] proposed the terminal guidance strategy of autonomous parafoil, which has good robustness. The trajectory of this method is the arc in the terminal planning strategy. Refs. [17,18] proposed the robust trajectory planning of parafoil in uncertain wind environments. Ref. [19] proposed a hybrid trajectory planning strategy, which is suitable for a class of parafoils. Using this method, the generated trajectory consists of multiple straight lines and arcs. Ref. [20] proposed a terminal trajectory planning method in the form of a Bézier curve. The path of this method is composed of straight lines and circles. Ref. [21] explored the direct multiple shooting method to obtain the optimal trajectories. In [22], parafoil guidance used the Line of Sight guidance algorithm. In [23], multiple parafoils coordinate and navigate autonomously according to the trajectory, which is a Dubins path composed of straight lines and arcs. For the multi system planning method, Refs. [24,25] explored the coordinated trajectory tracking method of multi parafoils to realize the accurate airdrop of multiple parafoils. Different from the traditional multi trajectory planning, the flight trajectory of the optimal trajectory planning is an irregular curve. Therefore, the optimal trajectory planning method can satisfy more complex constraints, such as terrain avoidance and multi-objective global optimization of control quantities [26–28]. For example, Refs. [29,30] applied multivariable control to the autonomous attitude optimal control system of parafoil. Meanwhile, unlike the fully autonomous airdrops, Ref. [31] explored a semi-autonomous Human-in-the-loop control method for precise landing. However, by analysing all the above research, we can observe that most planning methods have to recalculate the trajectory after changing the initial position or target position. Therefore, the lack of computing power will seriously limit the real-time performance of trajectory planning.

In recent years, deep reinforcement learning has been applied to autonomous trajectory optimization with its powerful learning ability. The deep reinforcement learning model is trained under the condition that the simulation environment and initial position are completely random. The simulation environment should include wind disturbance (including horizontal wind and vertical wind) and terrain change. Using the optimization model built by deep reinforcement learning, it can be launched at different initial positions without recalculating the flight trajectory. Based on this advantage, it is widely used in Unmanned Aerial Vehicles (UAVs) [32–36], automobile [37], and robots [38,39]. Ref. [40] explored an online path planning method based on deep reinforcement learning for UAV maneuvering target tracking and obstacle avoidance control. Ref. [41] explored an approach based on deep reinforcement learning to enable drones to perform navigation tasks in a multi-obstacle environment with randomness and dynamics. Ref. [42] explored a path planning method based on a deep neural network to solve the autonomous flight problem of quadrotor aircraft in unknown environments. In [43], an improved dual-delay deterministic strategy gradient is proposed for UAV energy-saving path planning. However, due to the monotonously decreasing height and fixed flight time of the parafoil system, it can be found from the above studies that ordinary UAV planning methods are not suitable for the parafoil system. Ref. [44] obtained a large number of training samples according to Kane equation (KE) and genetic algorithm (GA), and trained them by Back Propagation Neural Network (BPNN) by establishing the database of falling point trajectories. The trained neural network is used to calculate the trajectory parameters of airdrop under specific flight

conditions. However, this method needs to prepare a large number of parameter samples in advance for network training, and requires samples to be effective. In contrast, deep reinforcement learning does not need to build a sample database in advance. Furthermore, by analyzing the simulation results of the above research, although the traditional reinforcement learning method is improved in real time, the landing accuracy is not as good as the traditional numerical computation method.

Thus, we propose a real-time flight trajectory optimization method for a parafoil delivery system, which was based on an improved twin delayed deep deterministic policy gradient. First, based on the analysis of the actual flight data, combined with the flight environment and the characteristics of the parafoil, a 4-DOF model of the parafoil delivery system is built. Then, in the section of the optimization, we first introduce the principle of Deep Deterministic Policy Gradient (DDPG), explain the disadvantages of the DDPG algorithm, and introduce the improvement of Twin Delayed Deep Deterministic Policy Gradient (TD3) compared with DDPG. Based on the principle of TD3, the reason why the algorithm still leads to poor landing accuracy is explained. In order to reduce the impact of noise uncertainty on the exploration strategy and make better use of the good strategy of agent exploration, the existing TD3 method is improved by dynamically selecting the scale of noise in the action. By pre-evaluating the reward value of the action, different noise scales are selected according to the size of the pre-evaluated reward value. Simulation results show that the proposed method satisfies all trajectory constraints and shows the comparison results with the DDPG and TD3 algorithms. This method can achieve better real-time performance without losing landing accuracy.

## 2. Model of the Parafoil Delivery System

The parafoil delivery system is a very precise system, which has a unique structure. As shown in Figure 1, the parafoil consists of two parts: a flexible parafoil canopy and a payload. These two structures are connected by the lifting ropes and two controllable ropes. The controllable ropes are attached to the rear of the canopy. The shape of the canopy can be changed by manipulating the length of the controllable ropes. This phenomenon is so-called flap deflection. This method is used as the control input in the horizontal direction during the flight of the parafoil delivery system. With the flap deflection, the yaw angle of the parafoil delivery system can be controlled. Due to its good maneuverability, the parafoil delivery system has greatly improved the landing accuracy than the traditional parachute.

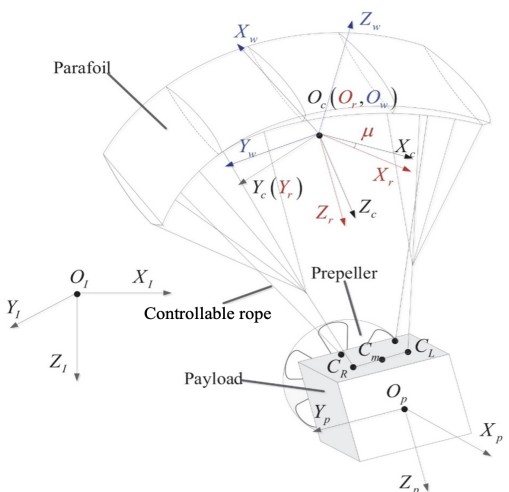

**Figure 1.** Parafoil delivery system.

The four main processes of flight tests are shown in Figure 2. The UAV system first carries the parafoil to the target altitude, and when it reaches the target altitude, the UAV releases it in the air. The whole flight process of the parafoil is controlled by the remote

control device. Data during flight, such as control quantities and yaw angles, are used to analyze the dynamics of parafoil. The physical parameters of the parafoil are shown in Table 1.

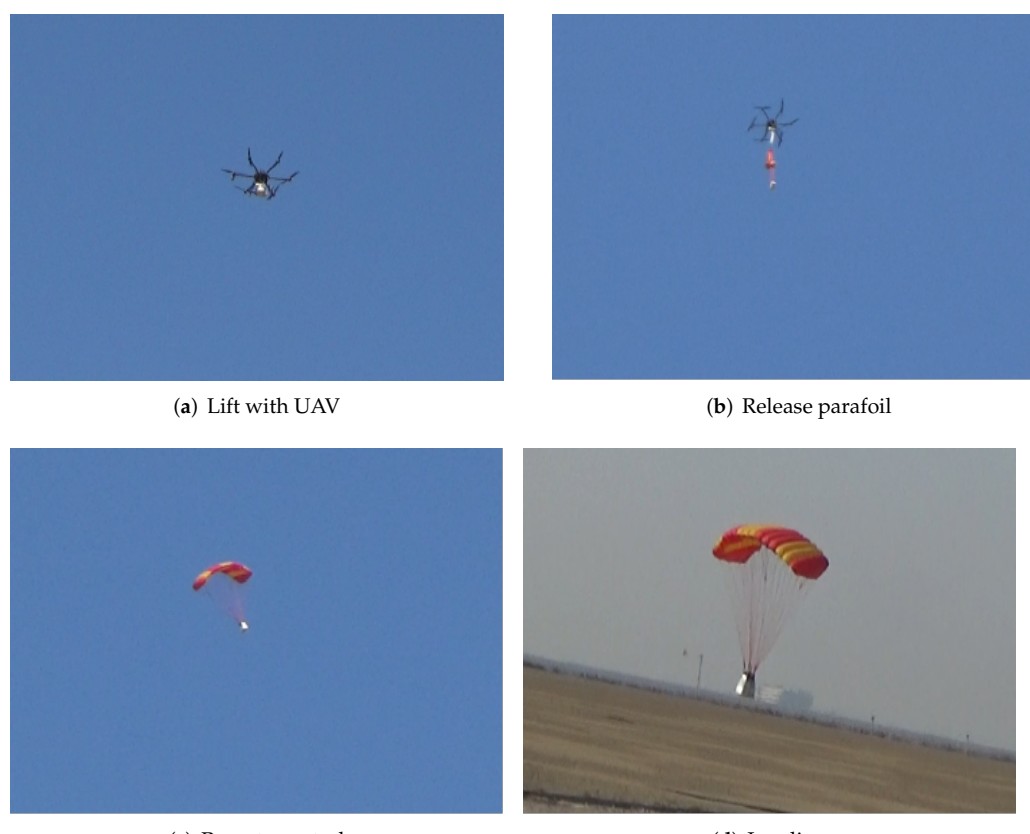

(**a**) Lift with UAV

(**b**) Release parafoil

(**c**) Remote control

(**d**) Landing

**Figure 2.** Process of flight test.

**Table 1.** Physical parameters of parafoil.

| Parameter | Value/Unit |
|:---:|:---:|
| Span | 2/m |
| Chord | 0.8/m |
| Area of canopy | $3/m^2$ |
| Length of suspending ropes | 1.4/m |
| Mass of payload | 12.5/kg |
| Mass of canopy | 0.9/kg |

Figure 3 records the data of a real flight trajectory of the parafoil. Figure 3a,b visually show the horizontal trajectory and 3D trajectory of the parafoil, respectively. The data in Figure 3c records the horizontal and vertical flight velocities of the parafoil at different times. The glide-ratio data are recorded in Figure 3d, and glide-ratio is defined as the ratio between the forward distance and the altitude drop during flight. The results show that, during the whole parafoil flight, the average vertical velocity and horizontal velocity are 3.7 m/s and 7.5 m/s, respectively. The average glide-ratio during flight is 2.1. The minimum turning radius of parafoil delivery system is 20 m. By analyzing the above real parafoil flight data, we built a 4-DOF model of parafoil:

$$\begin{cases} \dot{x} = v_{xy} \cos \varphi + v_{wx} \\ \dot{y} = v_{xy} \sin \varphi + v_{wy} \\ \dot{y} = v_z \\ \dot{\varphi} = \omega + u \end{cases} \tag{1}$$

where $[x, y, z]$ denotes the position of the parafoil system, $[v_{wx}, v_{wy}, v_{wz}]$ represents the component of wind velocity on the $x$-axis, $y$-axis, and $z$-axis, $v_{xy}$ denotes the horizontal velocity, and $v_z$ represents the vertical velocity of the parafoil system. In this work, it is assumed that the value of the component $v_{wz}$ of wind speed on the $z$-axis is equal to 0, so $z = v_z$. $\varphi$ represents the yaw angle of the parafoil system, $\dot{\varphi}$ is the angular velocity, $\omega$ represents the accumulated angular velocity of the parafoil, and $u$ represents the control input of parafoil system during flight, which is the angular acceleration.

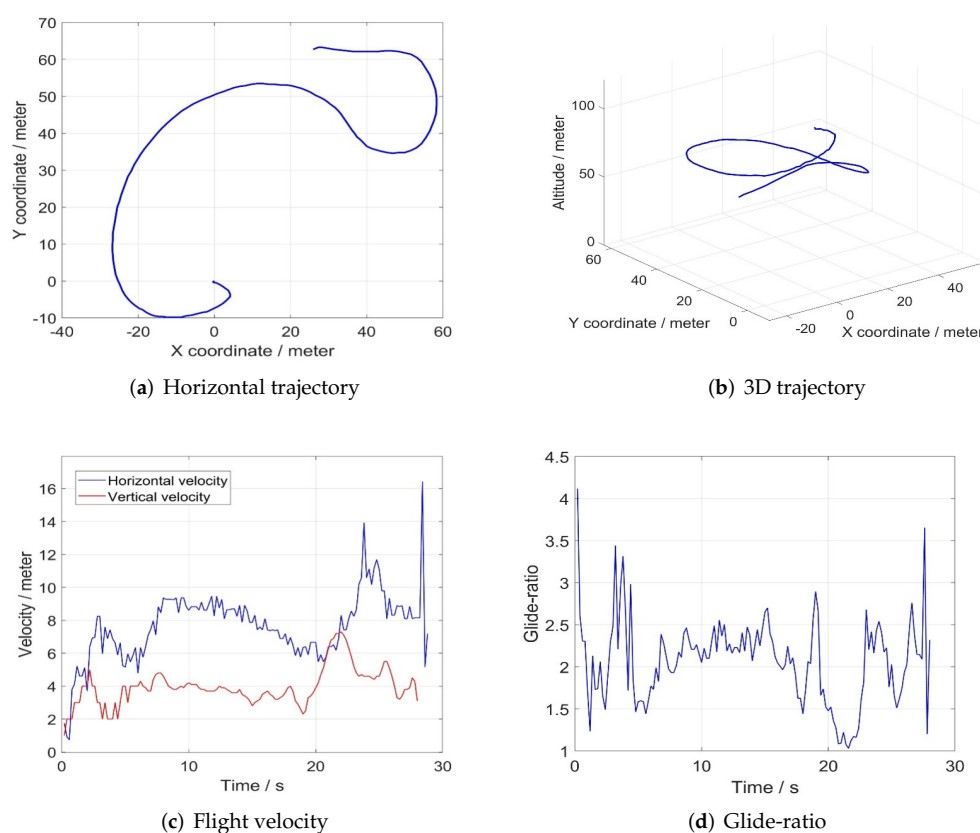

(**a**) Horizontal trajectory  (**b**) 3D trajectory

(**c**) Flight velocity  (**d**) Glide-ratio

**Figure 3.** Results of the flight test.

## 3. Trajectory Optimization Method Based on Improved Twin Delayed Deep Deterministic Policy Gradient

### 3.1. Deep Deterministic Policy Gradient Algorithm

The DDPG algorithm is proposed by the Google DeepMind team for realizing continuous action space control. It is composed of an Actor–Critic structure, combined with the Deep Q-learning Network (DQN) algorithm.

Figure 4 shows that the Actor–Critic structure consists of two parts: the actor network and the critic network. Among them, the actor represents the strategy network, which takes the current state as input and then generates the action under the current state through the analysis of neural networks. It takes advantage of Policy Gradient's ability to select actions in a continuous interval, and then selects actions randomly based on the learned action distribution. However, DDPG is different from Policy Gradient in that it generates deterministic actions based on the output of the actor, instead of generating according to Policy Gradient. The critic is the value network with a single step update. This update method solves the problem of low learning efficiency caused by the strategy gradient of

round update. Through the reward function to guide the learning direction of the network, the critic can obtain the potential rewards of the current state, and it takes the action output from the actor network as the input and outputs the evaluation value. Critic evaluates the action selected by the actor and guides the update direction of the network parameters of the actor, so that the actor after updating the network parameters can choose actions with a higher value as much as possible. The evaluation value Q is the reward for taking action $a_i$ under $S_i$. The formula is as follows:

$$Q(S_i, a_i | \theta^Q) \tag{2}$$

where $\theta^Q$ denotes the parameter of the critic network.

The behavior of each state directly obtains a certain action value through the deterministic policy function $\mu$:

$$a_t = \mu(S_t | \theta^\mu) \tag{3}$$

where $\mu$ represents the deterministic behavior policy, which is defined as a function and simulated by a neural network, $\theta^\mu$ represents the parameter of policy network, which is used to generate the determination action. $a_t$ is the change rate of angular velocity in state $S_t$, and the control input $u$ in (1) is changed by $a_t$.

In order to make the DDPG algorithm more random and learning coverage, it is necessary to add random noise to the selected action to make the value of the action fluctuate. The action after adding noise can be expressed as:

$$a_t \sim clip(N(\mu(S_t | \theta^\mu), \sigma^2), a_{low}, a_{high}) \tag{4}$$

where $N$ denotes the gaussian noise, and the noise follows the normal distribution, where $a_t$ is the expectation and $\sigma$ is the variance, $a_{low}$ is the minimum value of the action, and $a_{high}$ is the maximum value of the action.

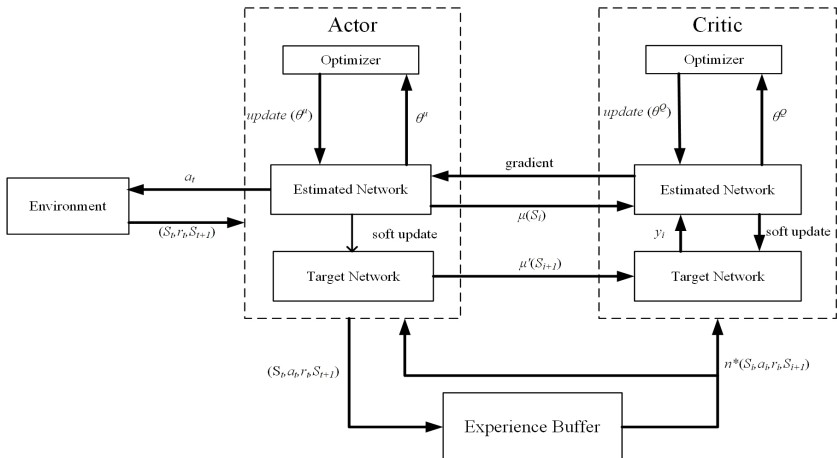

**Figure 4.** Schematic diagram of the Deep Deterministic Policy Gradient algorithm.

The design of DDPG is based on the off-policy approach, which separates the behavioral strategies from the evaluation strategies. There are estimated networks and target networks in the actor and the critic. Their estimated network parameters need to be trained, and the target network is soft-updated. Therefore, the two network structures of the actor and the critic are the same, but the network parameters are updated asynchronously. The soft update formula of the target network of the actor and the critic is as follows:

$$\begin{cases} \theta^{Q'} \leftarrow \tau\theta^Q + (1-\tau)\theta^{Q'} \\ \theta^{\mu'} \leftarrow \tau\theta^\mu + (1-\tau)\theta^{\mu'} \end{cases} \tag{5}$$

where $\tau$ represents the soft update rate, $\theta^Q$ and $\theta^\mu$ are the estimated network parameters of the actor and the critic, and $\theta^{Q'}$ and $\theta^{\mu'}$ are the target network parameters of the actor and the critic.

The action selected by the target network of the actor and the observation value of the environmental state are used as the input of the target network of the critic, which determines the update direction of the target network parameters of the critic. The update formula of the critic network parameters is:

$$y_i = r_i + \gamma Q'(S_{i+1}, \mu'(S_{i+1}|\theta^{\mu'}))|\theta^{Q'})) \tag{6}$$

$$L = \frac{1}{n}\sum_i^n (y_i - Q(S_i, a_i|\theta^Q))^2 \tag{7}$$

where $y_i$ represents the real evaluation value which is calculated by the target network, $S_i$ indicates environment status, $r_i$ represents the real reward, $a_i$ indicates the selected action under $S_i$, $\mu$ represents deterministic policy function, and $\gamma$ denotes the reward decay rate, which controls the influence of the reward value of the future step on the evaluation value of the current step. Larger $\gamma$ indicates that the critic pays more attention to future rewards, and smaller $\gamma$ indicates that the critic pays more attention to current rewards. $L$ denotes the loss function, which is the sum of squared errors between the actual value $y_i$ and the estimated value.

The update of the actor network parameters follows the deterministic strategy, whose formula is:

$$\nabla_{\theta^\mu} J = \frac{1}{n}\sum_i^n \nabla_a Q(S, a|\theta^Q)|_{S=S_i, a=\mu(S_i)} \nabla_{\theta^\mu} \mu(S|\theta^\mu)|_{S_i} \tag{8}$$

where $\nabla Q$ is from the critic, which is the update direction of the actor's network parameters, so that the actor with updated parameters can choose the action to obtain a higher evaluation value from the critic. $\nabla \mu$ is from the actor, which indicates the update direction of the parameters of the actor, so that the actor after updating the parameters is more likely to select the above action.

### 3.2. Improved Twin Delayed Deep Deterministic Policy Gradient Algorithm

Since DDPG is an off-policy method based on the DQN algorithm, each time it selects the highest value in the current state instead of using the actual action of the next interaction, there may be an overestimation. In the Actor–Critic framework of continuous action control, if each step is estimated in this way, the error will accumulate step by step, resulting in failure to find the optimal strategy and, ultimately, making the algorithm unable to converge. The twin delayed deep deterministic policy gradient (TD3) algorithm is optimized for mitigating the overestimation error of the DDPG algorithm.

The actor has two networks, an estimation network and a target network. The critic has two estimation networks and two target networks, respectively, as schematically illustrated in Figure 5. Thus, the critic has four networks with the same structure. The state quantity and action are the input of the critic network, and the output value is the value generated by the action executed in the current environment state. Regarding the optimization algorithm, TD3 adopts the Actor–Critic architecture similar to DDPG and is used to solve the problems in continuous action space. The improvement of the TD3 algorithm relative to the DDPG is mainly reflected in the following three aspects:

The first is the double critic network structure. In TD3, the critic's estimated network and target network have two, respectively. The smaller value of the target network is selected as the update target to update Estimated Critic1 and Estimated Critic2, which can alleviate the phenomenon of overestimation. TD3 uses the same method as DDPG to construct the loss function:

$$y_i = r_i + \gamma \min_{j=1,2} Q'(S_{i+1}, a_i'|\theta^{Qj'}) \tag{9}$$

$$L = \frac{1}{n} \sum_{i}^{n} (y_i - Q(S_i, a_i | \theta^Q))^2 \tag{10}$$

$S$ and $S_{i+1}$ are state quantities, as the input of the actor, and the output is the actions $a_i$ and $a_i'$ generated in the current environment.

The second is to delay updating the actor. In the TD3 algorithm, the critic network is updated once every step, the parameters of the actor are updated in a delayed manner, with a lower update frequency than the critic, that is, after the critic is updated multiple times, update the actor once. On the one hand, delaying updating the actor can reduce unnecessary repeated updates. On the other hand, it can also reduce errors accumulated in multiple updates.

The third is the smooth regularization of the target strategy. By adding noise based on the normal distribution to the action of the target network selection as (11):

$$a_i' = \mu'(S_{i+1} | \theta^{\mu'}) + clip(N(0, \sigma), -c, c) \tag{11}$$

the value function is updated more smoothly, the network is more robust, and the robustness of the algorithm is improved.

TD3 solves the problem of overvaluation of DDPG and facilitates the exploration of better strategies to improve the success rate and landing accuracy. However, applying the TD3 algorithm to the trajectory planning of the parafoil delivery system, combined with our existing simulation results and analyzing the experimental data, it is found that TD3 still has a larger landing error than the traditional trajectory optimization algorithm. This is difficult to solve only by increasing the number of training because the parafoil does not necessarily explore a better policy each time, or even a worse policy than the existing one, and stores it in the experience pool. This is due to the uncertainty of adding noise to the action. In the DDPG and the TD3 algorithms, in order to increase the randomness of the algorithm and the coverage of learning, they adopt the way of adding noise to the action to make it produce a certain fluctuation, hoping to explore more strategies. However, the action after increasing the noise is not necessarily better; it may make the action after increasing the noise obtain a lower reward value, thus storing a poor experience in the experience pool, which is not conducive to the algorithm learning a better strategy.

To solve this problem, we propose an improved twin delayed deep deterministic policy gradient algorithm, which dynamically changes the scale of noise to be added by evaluating the reward value of the selected action in advance. This method can effectively reduce the negative impact of noise uncertainty on strategy exploration, and make full use of excellent strategies.

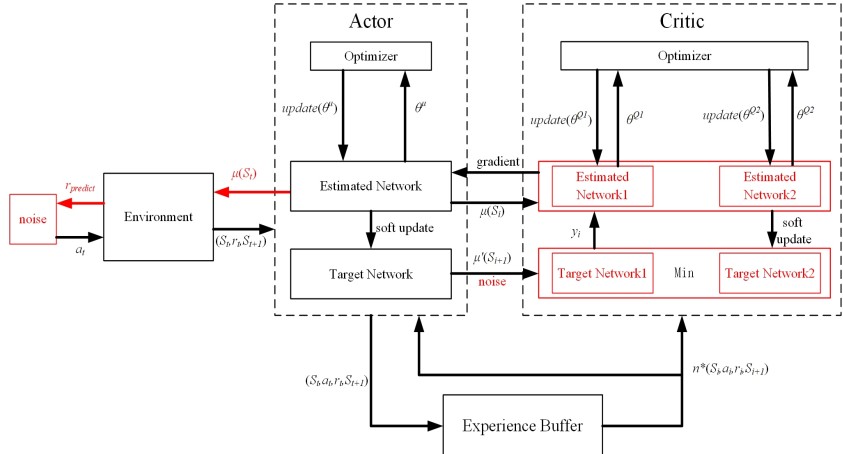

**Figure 5.** Schematic diagram of the Improved Twin Delayed Deep Deterministic Policy Gradient algorithm.

The action $\mu(S_t)$ selected in the $S_t$ state will first obtain the reward value $r_{predict}$ of environmental feedback without adding any noise, so as to pre-evaluate the value of the action. The purpose of pre-evaluation is to judge whether the action is an excellent strategy. If the action has a high value, reduce the scale of noise and maintain this strategy as much as possible. If the action has a low value, the scale of noise should be increased to explore better strategies. The variance $\sigma$ of Gaussian noise is determined by $r_{predict}$. The higher the reward $r_{predict}$ is, the smaller the variance $\sigma$ of noise is, the lower the reward $r_{predict}$ is, and the more the variance $\sigma$ of noise is. In this work, the maximum value of $\sigma$ is limited to 2, and the minimum value is 0.4. If the real-time reward value is less than 0, the exploration intensity should be increased, so $\sigma$ of negative action is 2. When the real-time reward value is greater than 0, $\sigma$ decreases along the positive direction of the $x$-axis, and the variation amplitude of $\sigma$ gradually decreases, but $\sigma$ cannot be reduced to 0, and a small amount of noise still needs to be retained to maintain the exploration. The agent uses action $a_t$ after adding noise to explore. This makes better use of the good strategies explored by the agents and, to some extent, avoids storing the worse explored steps in the experience pool. In (12), the scale of variance $\sigma$ is selected according to the pre-evaluated reward value. The improved TD3 algorithm is described in Algorithm 1, where the additional step 6 reflects our proposed modification:

$$
\sigma = \begin{cases} \frac{1}{0.8r_{predict}+0.5}, & if\ r_{predict} > 0 \\ 2, & otherwise \end{cases} \tag{12}
$$

---

**Algorithm 1** Improved TD3

---

1: Initial estimated critic network parameters $\theta^{Q1}, \theta^{Q2}$, and estimated actor network parameter $\theta^{\mu}$
2: Initial target networks parameters $\theta^{Q1'} \leftarrow \theta^{Q1}, \theta^{Q2'} \leftarrow \theta^{Q2}$
3: Set initial values of hyper-parameters according to the task requirements: experience playback buffer pool $B$, mini-batch size $n$, actor network learning rate $l_a$, critic network learning rate $l_c$, maximum episode $E$, soft update rate $\tau$
4: **for** $t = 1$ to $T$ **do**
5:     Select action $a_t = \mu(S_t|\theta^{\mu})$
6:     According to pre-evaluated reward value of action $a_t$, select the size of noise variance $\sigma$, and add noise $N$ to new action $a_t \sim clip(N(\mu(S_t|\theta^{\mu}), \sigma^2), a_{low}, a_{high})$ with noise, and observe reward $r_t$ in current state and new state $S_{t+1}$
7:     Store transition tuple $(S_t, a_t, r_t, S_{t+1})$ of this step in $B$
8:     Sample mini-batch of $n$ transactions $(S_i, a_i, r_i, S_{i+1})$ from $B$
9:     Compute target actions $a_i' = \mu'(S_{i+1}|\theta^{\mu'}) + clip(\mathcal{N}(0, \sigma), -c, c)$
10:     Compute Q-targets $y_i = r_i + \gamma \min_{j=1,2} Q'(S_{i+1}, a_i'|\theta^{Q'j})$
11:     Update estimated network parameters of the critic by minimizing loss: $L = \frac{1}{n}\sum_{i}^{n}(y_i - Q(S_i, a_i|\theta^{Q}))^2$
12:     **if** $t$ mod $d$ **then**
13:         Update the actor policy using sampled policy gradient:
14:         $\nabla_{\theta^{\mu}}J = \frac{1}{n}\sum_{i}^{n}\nabla_a Q(S, a|\theta^{Q})|_{S=S_i, a=\mu(S_i)}\nabla_{\theta^{\mu}}\mu(S|\theta^{\mu})|_{S_i}$
15:         Update parameters of target network of the critic and the actor:
16:         $\theta^{Qj'} \leftarrow \tau\theta^{Qj} + (1-\tau)\theta^{Qj'}$ for $j = 1,2$
17:         $\theta^{\mu'} \leftarrow \tau\theta^{\mu} + (1-\tau)\theta^{\mu'}$
18:     **end if**
19: **end for**

---

## 4. Trajectory Constraints and Reward Design

### 4.1. Trajectory Constraints

Trajectory constraints determine the optimization direction of the algorithm. According to the simulation environment of this work, the trajectory constraints include initial and terminal values. The initial value constraints of the parafoil delivery system are:

$$\begin{cases} x(t_0) = x_0 \\ y(t_0) = y_0 \\ z(t_0) = z_0 \\ \varphi(t_0) = \varphi_0 \end{cases} \tag{13}$$

where $t_0$ represents the initial time, and its value is 0, $[x_0, y_0, z_0]$ denotes the initial position, which is a random value, $\varphi_0$ denotes the initial yaw angle, which is also a random value. The terminal value constraints of parafoil delivery system are:

$$\begin{cases} t_f = \frac{z_0 - z_f}{v_z} \\ x(t_f) = x_f \\ y(t_f) = y_f \\ z(t_f) = z_f \end{cases} \tag{14}$$

where $t_f$ denotes the terminal time. In this work, $z_f = 0$ is assumed, so the value of $t_f$ is determined by the initial height $z_0$ and the vertical velocity $v_z$. It can be seen from (1) that the vertical velocity is constant, so the terminal time is a fixed time. $\left[ x_f, y_f \right]$ denotes the terminal location.

The parafoil in trajectory planning also needs to consider real-time path constraints, control input constraints, and terrain avoidance. In this article, these constraints will be considered in the reward design.

### 4.2. Reward Design

The system obtains rewards and punishments by interacting with the external environment. In this work, a positive value represents a reward and a negative value represents a punishment. First, each step of the whole flight must be rewarded and punished, so the horizontal difference between the remaining flying distance of the system and the distance to the destination should be calculated at each step, and consider the size of the control input. The optimization objective in the whole flight process is to minimize the horizontal error between the target position and the terminal position of the parafoil. Thus, the reward function is required to reflect the final landing result of the parafoil, and the terminal reward guides the parafoil to obtain higher landing accuracy. Then, the reward must consider the terrain avoidance. If the flight trajectory encounters a no-fly zone or exceeds the flyable area, a penalty will be given.

First, the setting of reward value should guide the optimization direction of each step in the trajectory planning process. The real-time trajectory constraints are:

$$D_t = \sqrt{(x_t - x_f)^2 + (y_t - y_f)^2} \tag{15}$$

$$d = \left| \frac{z_t}{v_z} \times v_{xy} - D_t \right| \tag{16}$$

$$r_u = 1 - |a_t| - \left| a_t - a_t' \right| \tag{17}$$

$$r_d = \begin{cases} 2, if\ d < 1\ and\ d' < 1 \\ 0.1 \times (d' - d), otherwise \end{cases} \tag{18}$$

$$r_t = r_d + 0.5 \times r_u \tag{19}$$

$D_t$ denotes the horizontal distance between the parafoil and the target point under $S_t$. $d$ and $d'$ are the difference between the remaining flying distance of the system and the distance to the destination under $S_t$ and $S_{t+1}$, respectively, $r_d$ represents the constraint of real-time distance under $S_t$, the range of $0.1 \times (d' - d)$ is $-1.5$ to $1.5$, so the maximum reward value when $d < 1$ and $d' < 1$ is set to 2. $r_u$ represents the constraint of real-time control input at $S_t$. Therefore, $r_t$ represents the reward of real-time constraints. In order to prevent the landing accuracy from decreasing due to excessive constraint control input, $r_u$ should be multiplied by the weight coefficient of 0.5 to appropriately reduce the weight. Under these real-time trajectory constraints, the optimization objective is to minimize the landing error in the whole flight process.

In addition, the parafoil will obtain a reward value $r_f$ in the terminal state, which is expressed as:

$$D_f = \sqrt{(x_{t_f} - x_f)^2 + (y_{t_f} - y_f)^2} \tag{20}$$

$$r_f = \begin{cases} \frac{K - D_f}{2}, t = t_f, D_f \leq 5 \\ \frac{K - D_f}{4}, t = t_f, D_f \leq 20 \\ \frac{K - D_f}{6}, t = t_f, D_f \leq 50 \\ M, t = t_f, D_f > 50 \end{cases} \tag{21}$$

where $\left[x_{t_f}, y_{t_f}\right]$ represents the final landing position of the parafoil, $D_f$ represents the landing error that is meant to be minimized, $K$ is a constant greater than 50, and $M$ is a negative constant. From (21), the formula shows that the value of $r_f$ increases with the decrease of landing error. The range of $r_t$ is $-1.5$ to $2.5$, so it is necessary to avoid diluting the terminal reward value $r_f$ by the $r_t$, the value of $K$ should be selected in combination with reward decay rate $\gamma$. When the landing error is different, the weight of the terminal reward value is different. When the landing error is less than 5 m, the weight of the terminal reward is the highest, and the parafoil pays more attention to the terminal reward value. When the landing error is large, the weight of the terminal reward value will be reduced, and the parafoil pays more attention to real-time reward values in order to obtain better planning strategies. If $D_f > 50$ indicates that the landing error is too large, the flight mission is deemed to have failed, and the reward value is M.

Finally, the reward function should also guide the parafoil to avoid multiple circular no-fly zones and limit the parafoil within the flying zone. Terrain avoidance reward function can be expressed as:

$$d_i = \sqrt{(x_t - x_i)^2 + (y_t - y_i)^2} \tag{22}$$

$$r_a = \begin{cases} M, d_i < radius \\ M, x_t < x_{min} \text{ or } x_t > x_{max} \text{ or } y_t < y_{min} \text{ or } y_t > y_{max} \end{cases} \tag{23}$$

where $[x_i, y_i]$ represents the central coordinate of no-fly zone, and *radius* represents the radius (minimum safe distance of parafoil) of the no-fly zone. $x_{min}$, $x_{max}$, $y_{min}$ and $y_{max}$ indicates the range of the flying region. The *M* indicates the penalty of flight failure. The higher the absolute value of negative number *M*, the greater the impact on this strategy, and the less likely the parafoil will adopt this strategy in the future. In (23), the formula indicates that, if the parafoil reaches the no-fly zone or exceeds the flying zone, it will be punished and the flight will be judged as a failure immediately.

## 5. Comparison

### 5.1. Simulation Environment

Figure 6 shows the main task space, which is a three-dimensional cube shaped region with a size of $500 \times 500 \times 500$ m. The initial point of the parafoil is initialized randomly in the region for each episode, and the red target point is set to [400,400,0]. The initial flying altitude of the parafoil is set at 500 m. There are three no-fly zones within the

flight area. They are cylinders with a radius of 50 m and a height of 500 m centered on [100,200], [250,400], and [300,100]. In addition, we also use another environment to test the performance of the improved TD3 algorithm in the wind disturbed environment. As shown in Figure 7, the centers of the three no-fly zones are [400,100], [200,200], and [300,400], respectively. The components of wind velocity in the positive direction of the *x*-axis, *y*-axis, and *z*-axis are −2 m/s, −1 m/s and 0 m/s, respectively.

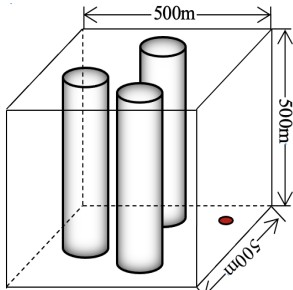

**Figure 6.** The simulation environment 1.

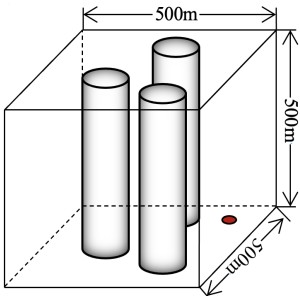

**Figure 7.** The simulation environment 2.

*5.2. Training and Results*

The DDPG, TD3 and improved TD3 algorithms have the same parameters during training. Table 2 shows the main super parameters of the three algorithms. The max episodes of the algorithm is 3000, which represents the maximum number of episodes of the three algorithms. $\gamma$ ranges from 0 to 1, the larger the value of $\gamma$, the greater the impact of the reward value of future steps on the current state. $\tau$ is the soft update factor in (5). The learning rate setting should not be too large or too small to avoid failing to converge or falling into local optimization. Reply buffer size represents the maximum number of experiences that can be stored; when the reply buffer is full, the newly stored experience will replace the earliest experience. Batch size is the number of samples taken from the reply buffer during each training. In this work, the value of delay step is 3, which means that the parameter of the actor is updated only once after the parameter of the critic is updated three times. As shown in Figure 8, the initial point of each episode is initialized randomly in the task area to ensure that the training process covers as many positions as possible. The episode ends when the parafoil flight time ends or the parafoil touches the restricted zone or crosses the boundary. DDPG, TD3, and the improved TD3 algorithm are used to train the parafoil delivery system in the task area. Both the actor and the critic networks use the multilayer perceptron (MLP) network structure. Figure 9 shows the changes of the average reward values of the three algorithms during the training of 3000 episodes in simulation environment 1.

**Table 2.** Parameters of the training experiment.

| Parameter | Value |
|---|---|
| Max Episodes | 3000 |
| Discount Factor $\gamma$ | 0.99 |
| Soft Update Factor $\tau$ | 0.01 |
| Critic Learning Rate | 0.0005 |
| Actor Leaning Rate | 0.001 |
| Reply Buffer Size | $5 \times 10^5$ |
| Batch Size | 256 |
| Delay Steps | 3 |

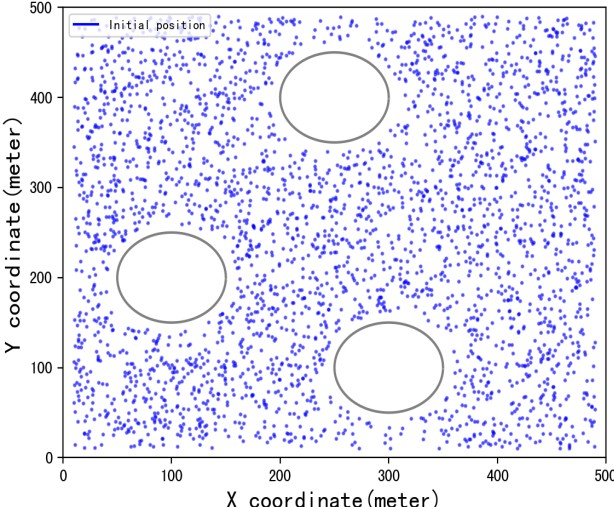

**Figure 8.** The initial positions of 3000 episodes.

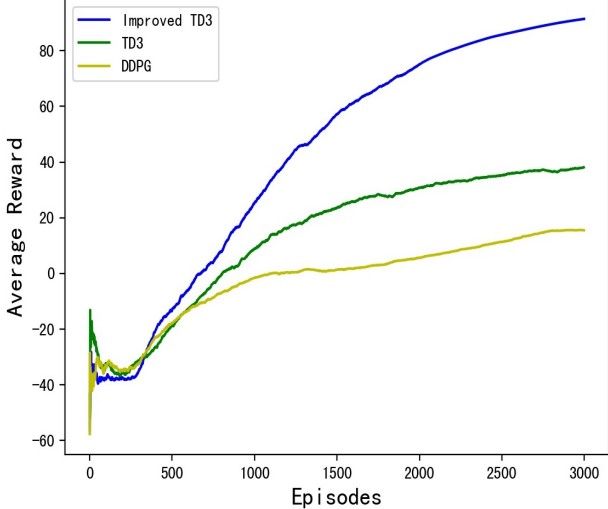

**Figure 9.** Average reward value of 3000 episodes.

As shown in Figure 9, the parafoil has not learned excellent strategies at the initial stage, so the probability of touching the no-fly zone or exceeding the flying region is high. It can be seen that the average return of the parafoil random exploration in the environment is low, usually a negative reward value. When the playback buffer size reaches a certain amount, the parafoil starts to train the network to update the policy. Using the method in this work, the noise to be added is selected according to the pre-evaluated reward value after the action is selected, and the noise decreases with the increase of times. It can be

observed that, with the increase of episodes, the parafoil delivery system gradually learns excellent strategies for trajectory planning, so the average reward value gradually increases. The model trained by neural network can cover all the initial positions in the environment.

### 5.3. Testing and Results

Five indicators were used to evaluate the training results, including landing errors LR10, LR20 and LR50, crash rate, and average landing errors. LR10, LR20, and LR50 represent the percentage of episodes that the distance between the final landing point of the parafoil, and the target point is less than 10 m, 20 m and 50 m after 100 tests, respectively. Crash rate is the percentage of times the parafoil hits the no-fly zone or exceeds the flying zone after 100 tests. The average landing error is the sum of the total errors of successful landing divided by the number of successful landings. The lower the crash rate is, the higher the success rate of the model is. The lower the average landing error is, the higher the landing accuracy of the model is. In the simulation environment 1 shown in Figure 6, the test results of DDPG, TD3, and improved TD3 after testing 100 episodes are shown in Table 3. In the simulation environment 2 shown in Figure 7, the test results of DDPG, TD3, and improved TD3 after testing 100 episodes are shown in Table 4.

**Table 3.** Results of 100 tests using DDPG, TD3, and improved TD3 in the simulation environment 1.

| Algorithm | LR10 | LR20 | LR50 | Crash | Average Landing Errors |
|---|---|---|---|---|---|
| DDPG | 1% | 19% | 63% | 20% | 33.8 m |
| TD3 | 17% | 74% | 92% | 13% | 16.6 m |
| Improved TD3 | 89% | 92% | 93% | 5% | 5.7 m |

**Table 4.** Results of 100 tests using DDPG, TD3, and improved TD3.

| Algorithm | LR10 | LR20 | LR50 | Crash | Average Landing Errors |
|---|---|---|---|---|---|
| DDPG | 5% | 41% | 60% | 33% | 37.7 m |
| TD3 | 8% | 48% | 82% | 15% | 21.8 m |
| Improved TD3 | 76% | 91% | 94% | 4% | 6.4 m |

It can be seen from Table 3 that the average landing error of the improved TD3 algorithm is 5.7 m, and the value of LR10 is 89%, which is much higher than the other two algorithms. The average landing error of the DDPG algorithm and TD3 algorithm are 33.8 m and 16.6 m. Thus, the improved TD3 algorithm has the highest landing accuracy. In addition, the crash rate of the improved TD3 algorithm is only 5%, while the crash rates of the DDPG algorithm and TD3 algorithm are 20% and 13%, respectively. Thus, the improved TD3 algorithm also has the highest success rate and the lowest crash rate. In Table 4, the improved TD3 algorithm also has the highest landing accuracy and success rate.

Figures 10–13 show four typical cases of successfully reaching the target point using the improved TD3 algorithm in simulation environment 1. Their initial positions were [150,150], [100,400], [250,250], and [400,130], and the initial height is 500 m. Parafoil landing positions are [399.2,395.3], [401.8,395.6], [399.6,395], and [402.7,404.2]. The flight time of parafoil is determined by its initial altitude, which is 150 s. The range of control input is −1 to 1. The yaw angle is recorded as 0° to 360°. From these cases, we can see that the parafoil delivery system has high landing accuracy, and the fluctuation of control input and the value of control input have also been controlled. From the reference trajectory, it can be seen that the parafoil has learned excellent strategies. When $d$ is small and the remaining flight time is large, the parafoil can hover in the air and consume altitude.

Figures 14–17 show four typical cases of successfully reaching the target point using the improved TD3 algorithm in simulation environment 2. Their initial positions are randomly selected as [198,432], [115,234], [232,56], and [338,456], and the initial height is 500 m. Parafoil landing positions are [403.4,402.6], [401.1,401.5], [399,400.9], and [403.6,399].

Different from simulation environment 1, the horizontal velocity of parafoil will change in different directions due to the influence of wind velocity. If the horizontal velocity decreases, the parafoil may not have redundant height. The control input needs to ensure that the parafoil flies directly to the target point as far as possible without consuming height. This is the reason why the control input in simulation environment 2 fluctuates frequently, but the improved TD3 algorithm can still constrain the size of the control input to make the control input as small as possible.

The experimental results of the three algorithms under random initial points are also compared. Figures 18–20 show the initial positions of parafoil under simulation environment 1 as [439,299], [72,312], and [339,452], DDPG, TD3, and improved TD3 were used for testing, respectively. Figures 21 and 22 show the initial positions of parafoil in simulation environment 2 as [432,325] and [165,99]. The landing errors of the three algorithms are shown in Table 5. It can be seen from Figure 19c that, because the learning ability of the DDPG is poor compared with the other two algorithms, it can not learn a better input control strategy, resulting in a high input control oscillation frequency of the DDPG. The oscillations in Figure 19d are due to the fact that 360° and 0° are considered equal in this work. When the initial positions are [439,299] and [72,312], the landing accuracy of improved TD3 is obviously better than DDPG and TD3. When the initial position is [339,452], although the landing accuracy of the improved TD3 is slightly worse than that of TD3, it can be seen from Figure 20 that the trajectory of the improved TD3 algorithm is more consistent with the setting of the reward function and can effectively consume altitude, and the control input value is smaller, thus proving that the training effect of the improved TD3 is better.

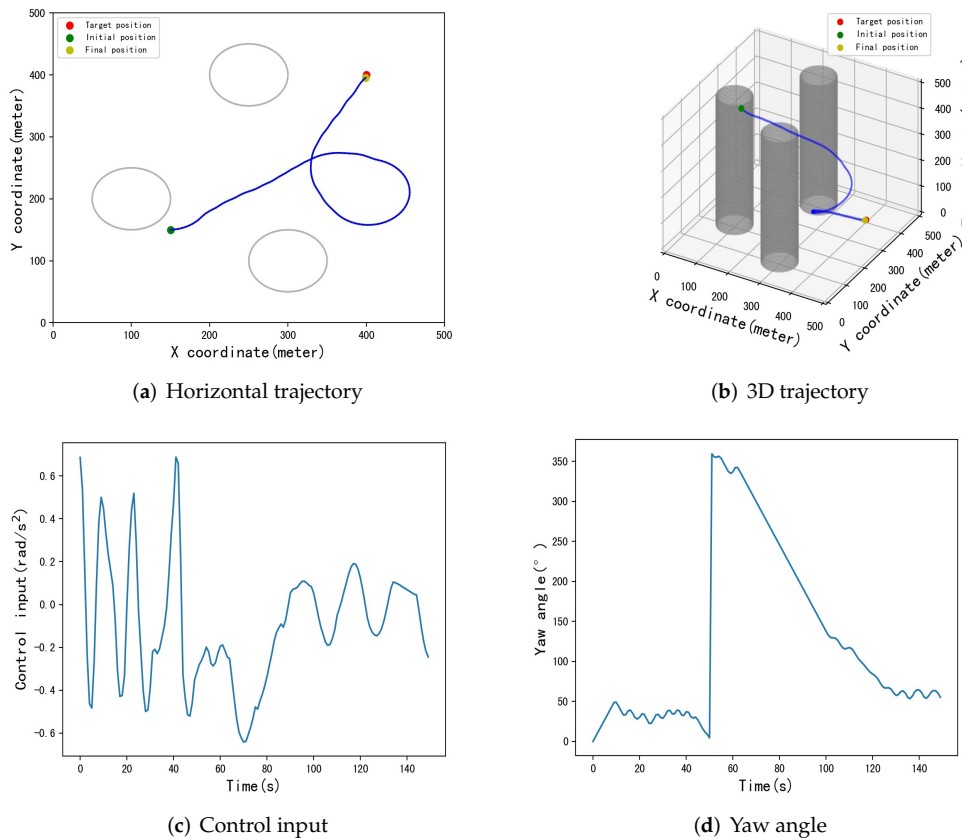

(**a**) Horizontal trajectory

(**b**) 3D trajectory

(**c**) Control input

(**d**) Yaw angle

**Figure 10.** Improved TD3-Case1 in simulation environment 1.

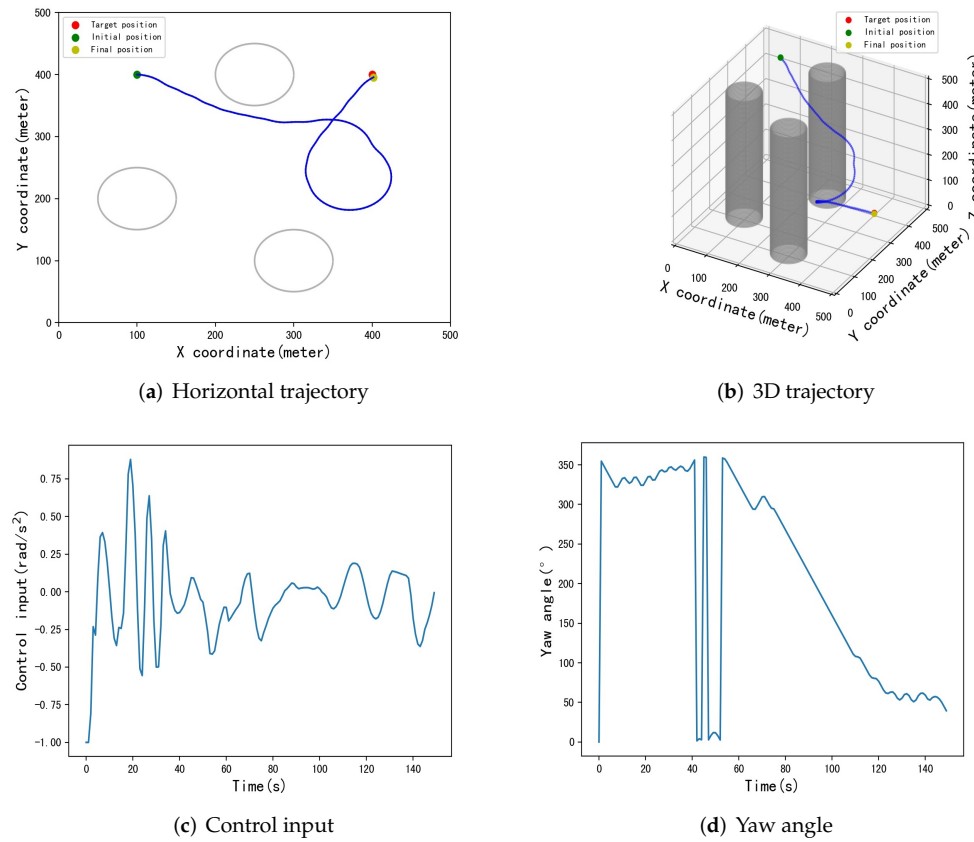

**Figure 11.** Improved TD3-Case2 in simulation environment 1.

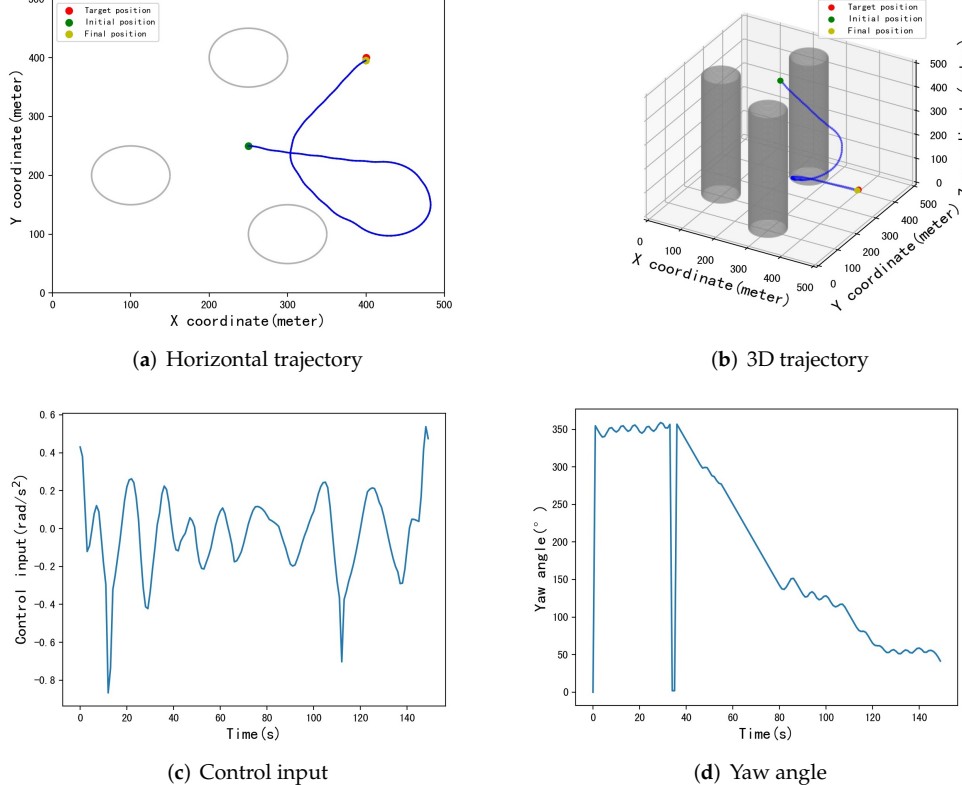

**Figure 12.** Improved TD3-Case3 in simulation environment 1.

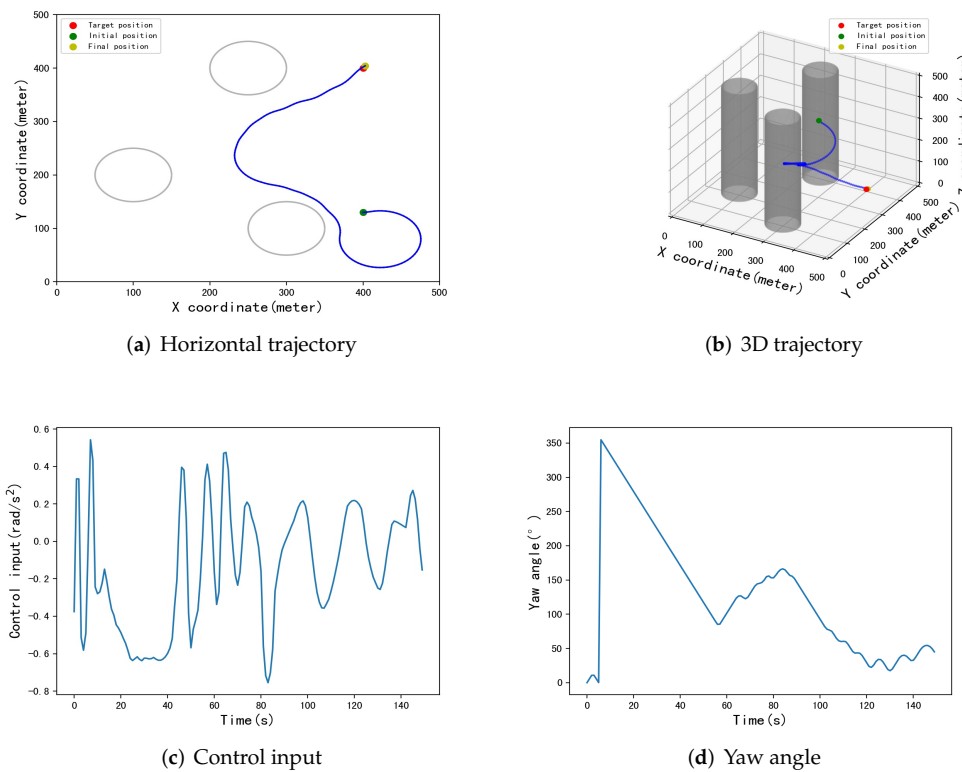

(**a**) Horizontal trajectory

(**b**) 3D trajectory

(**c**) Control input

(**d**) Yaw angle

**Figure 13.** Improved TD3-Case4 in simulation environment 1.

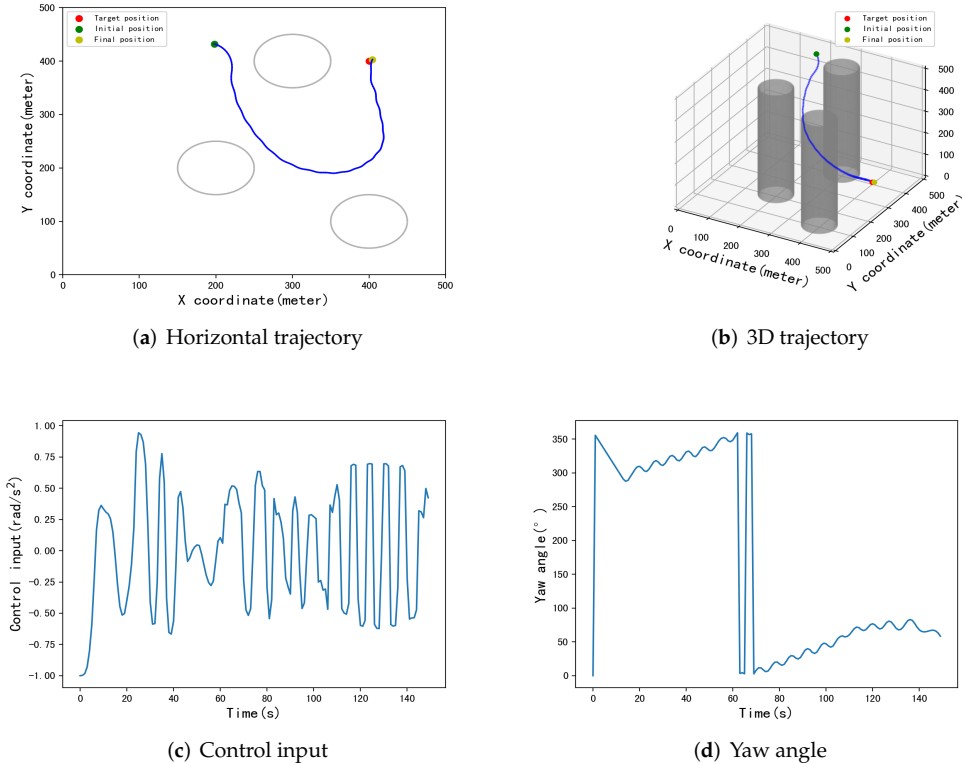

(**a**) Horizontal trajectory

(**b**) 3D trajectory

(**c**) Control input

(**d**) Yaw angle

**Figure 14.** Improved TD3-Case1 in simulation environment 2.

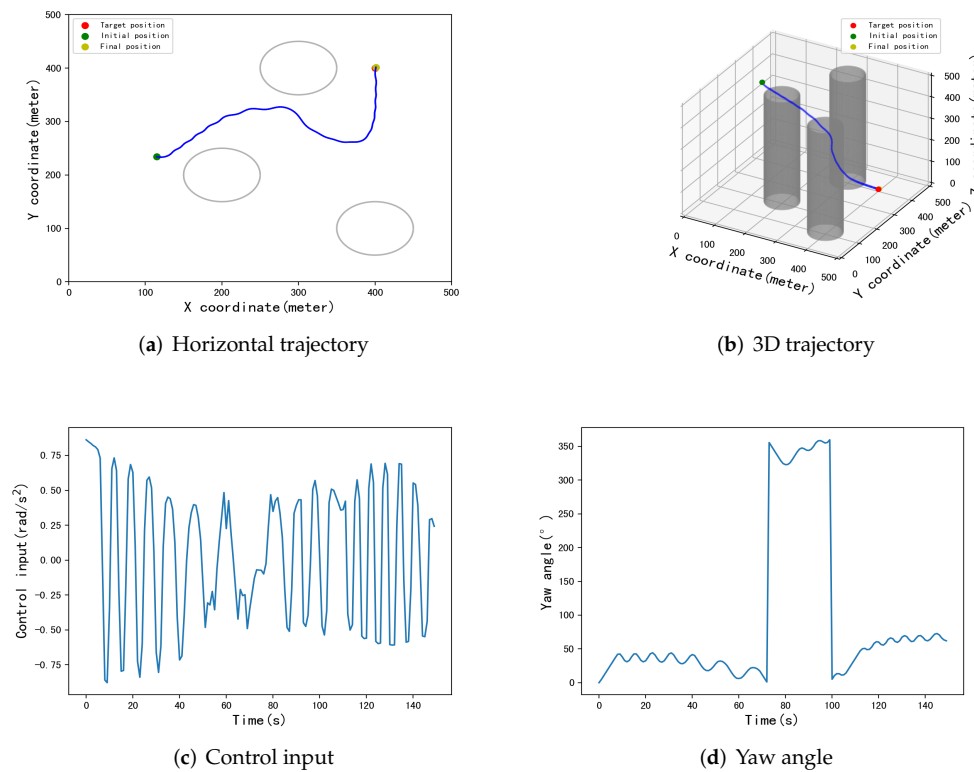

(**a**) Horizontal trajectory

(**b**) 3D trajectory

(**c**) Control input

(**d**) Yaw angle

**Figure 15.** Improved TD3-Case2 in simulation environment 2.

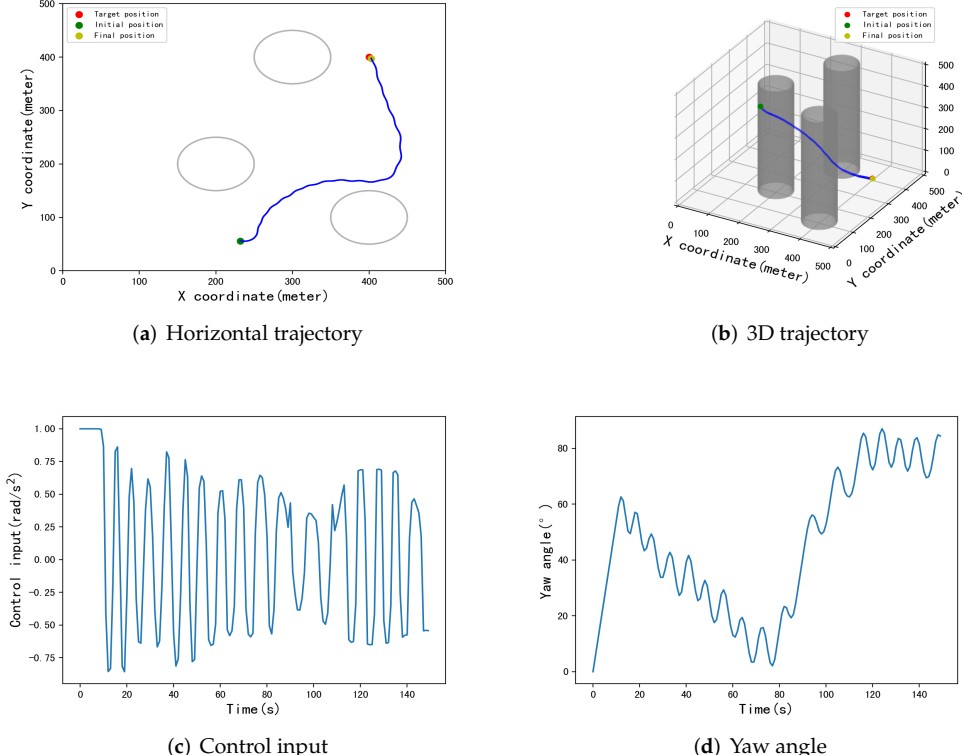

(**a**) Horizontal trajectory

(**b**) 3D trajectory

(**c**) Control input

(**d**) Yaw angle

**Figure 16.** Improved TD3-Case3 in simulation environment 2.

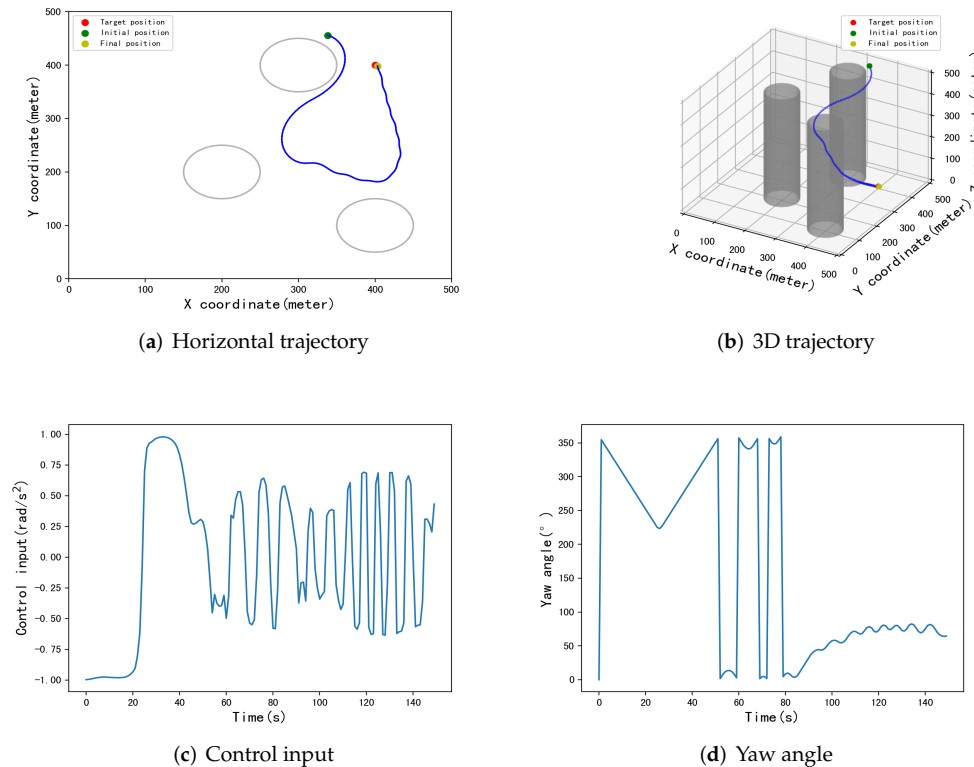

(**a**) Horizontal trajectory

(**b**) 3D trajectory

(**c**) Control input

(**d**) Yaw angle

**Figure 17.** Improved TD3-Case4 in simulation environment 2.

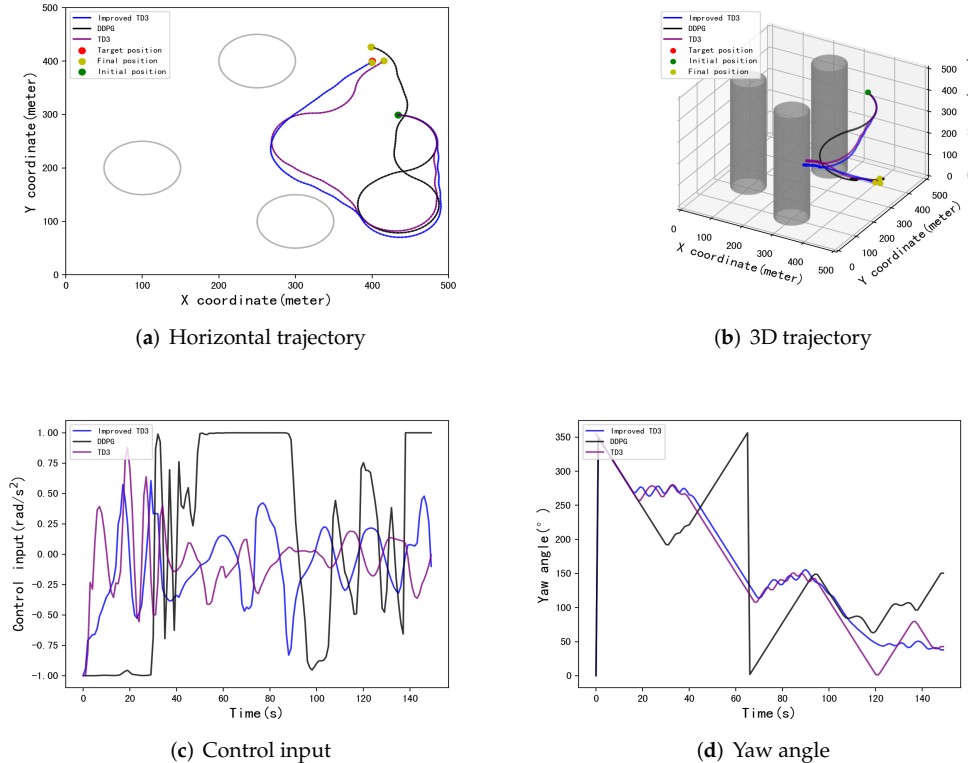

(**a**) Horizontal trajectory

(**b**) 3D trajectory

(**c**) Control input

(**d**) Yaw angle

**Figure 18.** Comparison results of three algorithms when the initial point is [434,299].

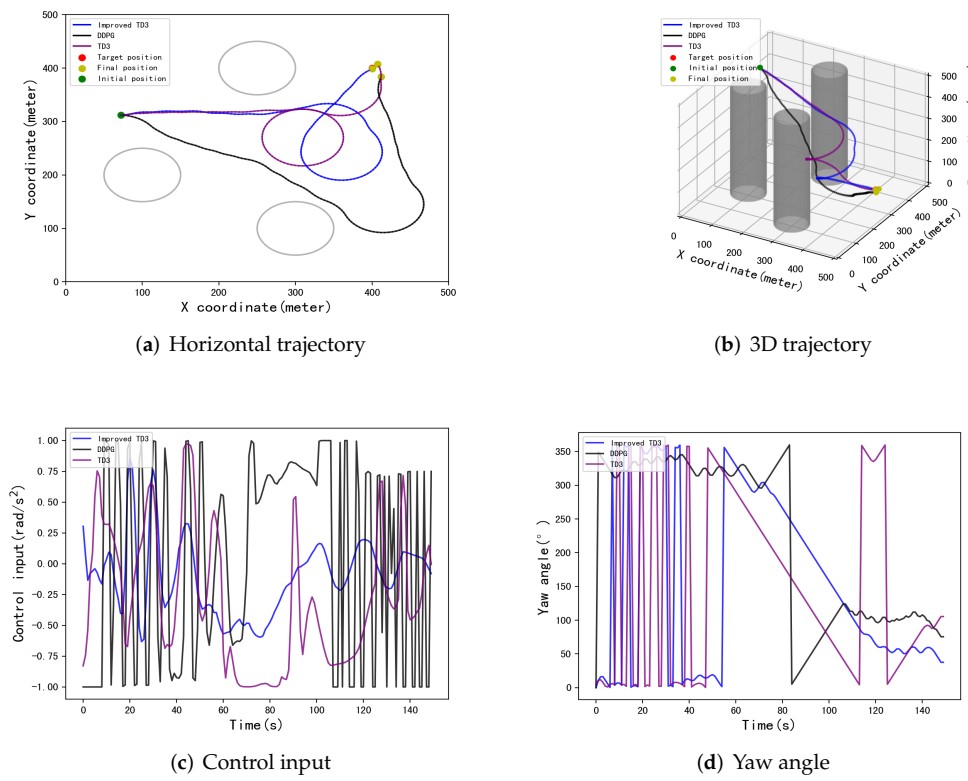

(**a**) Horizontal trajectory

(**b**) 3D trajectory

(**c**) Control input

(**d**) Yaw angle

**Figure 19.** Comparison results of three algorithms when the initial point is [72,312].

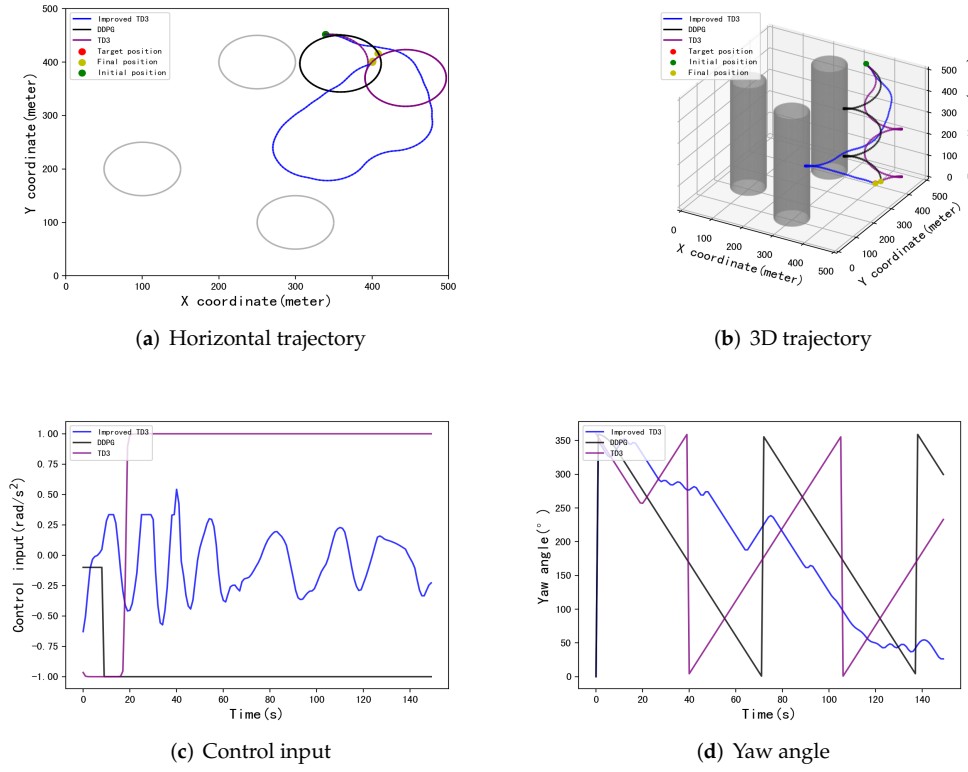

(**a**) Horizontal trajectory

(**b**) 3D trajectory

(**c**) Control input

(**d**) Yaw angle

**Figure 20.** Comparison results of three algorithms when the initial point is [339,452].

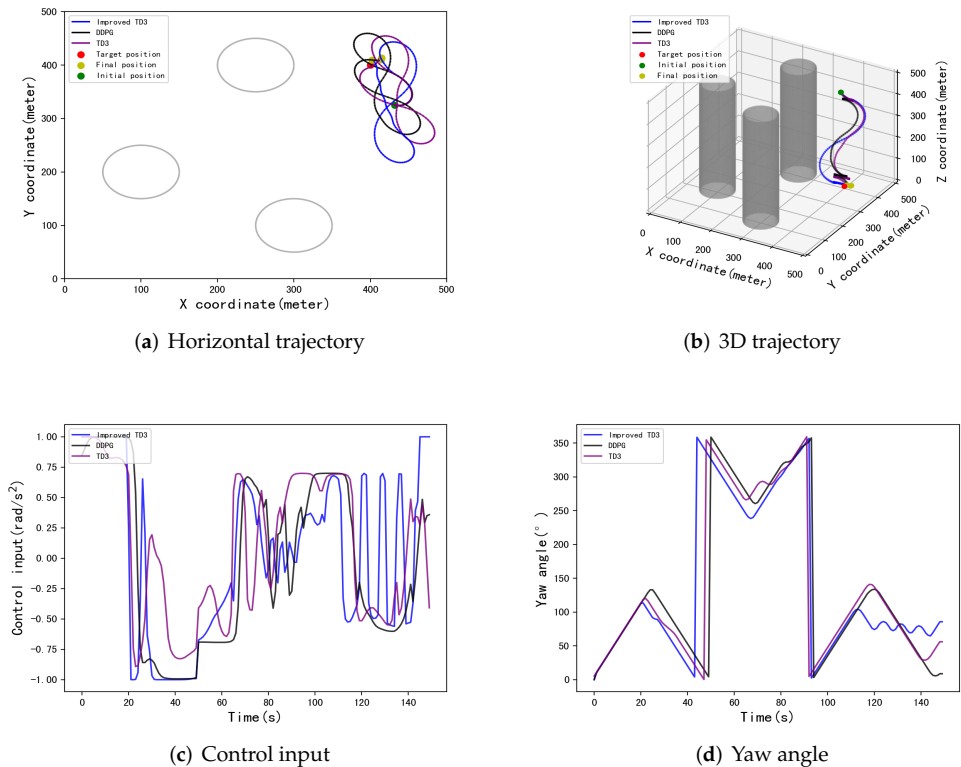

(**a**) Horizontal trajectory      (**b**) 3D trajectory

(**c**) Control input      (**d**) Yaw angle

**Figure 21.** Comparison results of three algorithms when the initial point is [432,325].

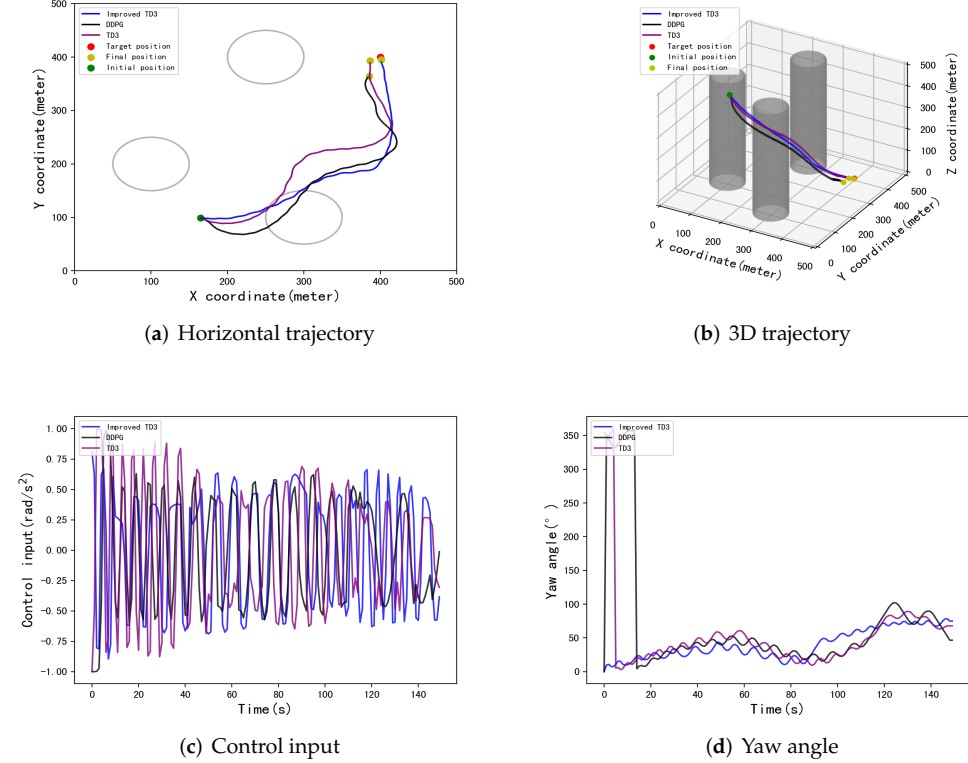

(**a**) Horizontal trajectory      (**b**) 3D trajectory

(**c**) Control input      (**d**) Yaw angle

**Figure 22.** Comparison results of three algorithms when the initial point is [165,99].

**Table 5.** Results of three cases using DDPG, TD3, and improved TD3.

| Initial Position | DDPG | | TD3 | | Improved TD3 | |
| --- | --- | --- | --- | --- | --- | --- |
| | Final Position | Landing Error | Final Position | Landing Error | Final Position | Landing Error |
| [434,299] | [398.8,426.7] | 26.7 m | [415.4,400.5] | 15.4 m | [399.7,397.6] | 2.4 m |
| [72,312] | [412,383.9] | 20 m | [407.4,407.5] | 10.5 m | [400.3,398.4] | 1.6 m |
| [339,452] | [408,416.5] | 18.3 m | [400,400.1] | 0.1 m | [400.9,397.8] | 2.4 m |
| [432,325] | [414.2,408.5] | 16.5 m | [415.4,414.7] | 21.2 m | [402,409.6] | 9.8 m |
| [165,99] | [385.6,364.4] | 38.4 m | [386.6,393] | 15.1 m | [401.6,396.5] | 3.8 m |

In addition, Figure 23 shows that, in simulation environment 1, when the initial point is the extreme case of [20,180], receiving strong interference from the terrain, all three algorithms fail. Figure 24 shows that, when the initial position is [120,440], the improved TD3 algorithm succeeds, but the DDPG and the TD3 fail.

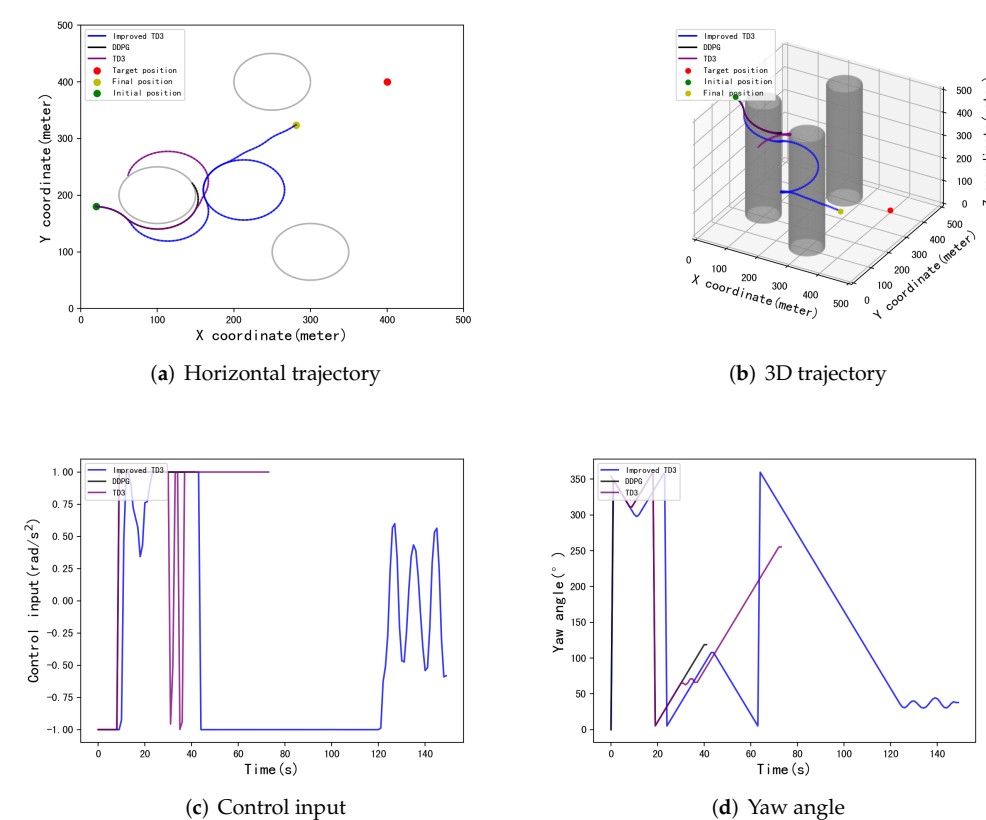

(**a**) Horizontal trajectory

(**b**) 3D trajectory

(**c**) Control input

(**d**) Yaw angle

**Figure 23.** All three algorithms fail.

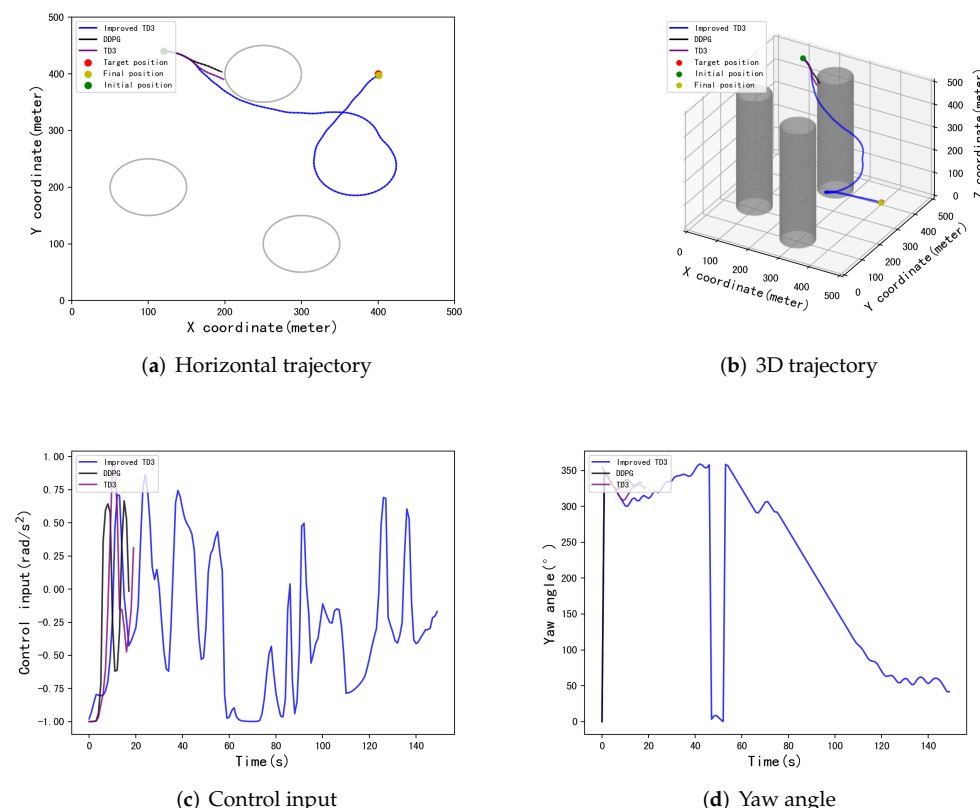

(**a**) Horizontal trajectory

(**b**) 3D trajectory

(**c**) Control input

(**d**) Yaw angle

**Figure 24.** Only the improved TD3 algorithm is successful.

## 6. Conclusions

In this work, a trajectory planning method based on deep reinforcement learning is proposed, which enables the parafoil to meet the autonomous trajectory planning under complex constraints. It proposes a new method of selecting noise according to the pre-evaluation reward value to improve the Twin Delayed Deep Deterministic Policy Gradient algorithm. It solves the problem in which the agent explores poor strategies due to the randomness of noise. Firstly, by analyzing the actual flight data of the parafoil system, combined with the mission environment and the characteristics of the parafoil, a 4-DOF model is built. Then, based on the Improved Twin Delayed Deep Deterministic Policy Gradient, the parafoil trajectory planning method is described in detail. Simulation results show that the Improved Twin Delayed Deep Deterministic Policy Gradient algorithm can realize trajectory planning at different initial positions, and the landing accuracy and success rate are significantly improved compared with Deep Deterministic Policy Gradient and Twin Delayed Deep Deterministic Policy Gradient. The method in this paper improves the negative impact of noise uncertainty on exploration; due to the randomness of the extraction experience, the bad experience that has been stored in the reply buffer may still be extracted and learned. The next work can use the experience of the first playback mechanism to improve this defect.

**Author Contributions:** Conceptualization, J.Y., H.S. and J.S.; Formal analysis, J.Y.; Investigation, J.Y.; Methodology, J.Y. and H.S.; Writing—original draft, J.Y.; Writing—review & editing, H.S. and J.S. All authors have read and agreed to the published version of the manuscript.

**Funding:** National Natural Science Foundation of China: 62003177, 62003175 and 61973172.

**Institutional Review Board Statement:** Not applicable.

**Informed Consent Statement:** Not applicable.

**Data Availability Statement:** Not applicable.

**Conflicts of Interest:** The authors declare no conflict of interest.

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
