# Peer review of "Improved Twin Delayed Deep Deterministic Policy Gradient Algorithm Based Real-Time Trajectory Planning for Parafoil under Complicated Constraints"

_applsci, doi:10.3390/app12168189_

Round 1

Author Response

First and foremost, we thank the reviewer and editors for giving us such valuable advises and useful comments about this manuscript. These comments have been very helpful for improving our manuscript and pointing out some future research directions. We have addressed all the comments carefully, and we have made extensive amendments accordingly.

In the document, we highlighted the comments of the reviewer using blue font in this document. The responses to these comments are highlighted by black font. The revisions in the manuscript are highlighted by red font and shown in the purple frame of the document.

Reviewer 2 Report

My recommendations for improving the article:

Missing space between text and references, for example in lines 28 and 53.

In line 71, after the text wind disturbance, I recommend adding: (including horizontal and vertical gusts)

In the text, for example, in lines 110, 111 and 112, you use the term controllable ropes. But in Figure 1 you used the term control line. I recommend using the term controllable rope in Figure 1 as well.

Table 1 shows the data 12.8m for Length of suspending ropes. I think that is the wrong figure and it is too much for a parafoil with a span of 2 meters, chord 0.8 meters and an area of canopy of 3 square meters.

In line 259, I recommend adding units for the size of simulation environment: 500 x 500 x 500 meters.

Author Response

(The authors gave the same response as above.)
